# Cohort profile: Worldwide Collaboration on OsteoArthritis prediCtion for the Hip (World COACH) – an international consortium of prospective cohort studies with individual participant data on hip osteoarthritis

Michiel M A van Buuren [1] Noortje S Riedstra,[1] Myrthe A van den Berg,[1] Fleur D E M Boel,[1] Harbeer Ahedi,[2] Vahid Arbabi,[3,4] Nigel K Arden,[5] Sita M A Bierma-Zeinstra,[6] Cindy G Boer,[7] Flavia Cicuttini [8] Timothy F Cootes,[9] Kay Crossley,[10] David Felson [11] Willem Paul Gielis,[3,12] Joshua Heerey,[10] Graeme Jones,[2] Stefan Kluzek,[5] Nancy E Lane,[13] Claudia Lindner,[9] John A Lynch [14] J Van Meurs,[1,7] Andrea B Mosler,[10] Amanda E Nelson,[15] M Nevitt,[14] Edwin Oei,[16] Jos Runhaar,[6] Jinchi Tang,[1] Harrie Weinans,[3,17] Rintje Agricola[1]

For numbered affiliations see end of article.

**Correspondence to**
Dr Harrie Weinans;
H.H.Weinans@umcutrecht.nl and
Dr Rintje Agricola;
r.agricola@erasmusmc.nl

## ABSTRACT

**Purpose** Hip osteoarthritis (OA) is a major cause of pain and disability worldwide. Lack of effective therapies may reflect poor knowledge on its aetiology and risk factors, and result in the management of end-stage hip OA with costly joint replacement. The Worldwide Collaboration on OsteoArthritis prediCtion for the Hip (World COACH) consortium was established to pool and harmonise individual participant data from prospective cohort studies. The consortium aims to better understand determinants and risk factors for the development and progression of hip OA, to optimise and automate methods for (imaging) analysis, and to develop a personalised prediction model for hip OA.

**Participants** World COACH aimed to include participants of prospective cohort studies with ≥200 participants, that have hip imaging data available from at least 2 time points at least 4 years apart. All individual participant data, including clinical data, imaging (data), biochemical markers, questionnaires and genetic data, were collected and pooled into a single, individual-level database.

**Findings to date** World COACH currently consists of 9 cohorts, with 38 021 participants aged 18–80 years at baseline. Overall, 71% of the participants were women and mean baseline age was 65.3±8.6 years. Over 34 000 participants had baseline pelvic radiographs available, and over 22 000 had an additional pelvic radiograph after 8–12 years of follow-up. Even longer radiographic follow-up (15–25 years) is available for over 6000 of these participants.

**Future plans** The World COACH consortium offers unique opportunities for studies on the relationship between determinants/risk factors and the development or progression of hip OA, by using harmonised data on clinical findings, imaging, biomarkers, genetics and lifestyle. This provides a unique opportunity to develop a personalised hip OA risk prediction model and to optimise methods for imaging analysis of the hip.

### STRENGTHS AND LIMITATIONS OF THIS STUDY

⇒ The Worldwide Collaboration on OsteoArthritis prediCtion for the Hip (World COACH) consortium brings patients together with a highly qualified and multidisciplinary team of experts and young investigators in the field of hip osteoarthritis, with backgrounds in orthopaedic surgery, rheumatology, physical therapy, general practice, genetics, epidemiology, biostatistics, technical medicine, biomechanical engineering, radiology, imaging science and artificial intelligence.

⇒ The World COACH consortium is unique for having harmonised individual participant data on clinical measurements, radiological imaging, biochemical markers, lifestyle and diet, comorbidities, medication, physical and cognitive functioning, quality of life and genetics from over 38 000 people, both from the general population as well as from specific populations at risk for hip osteoarthritis.

⇒ The World COACH consortium has sequential hip radiography available for each participant with a follow-up duration ranging from 5 to over 25 years.

⇒ The main limitations of the consortium are the geographical origins of the included cohorts (Western world) and the heterogeneity in collected data by the cohorts, which may limit the possibilities of harmonisation.



## INTRODUCTION

Osteoarthritis (OA) is a common disease and a leading cause of disability in adults.[1] Over 500 million people are affected by OA worldwide, leading to a global prevalence of around 7%.[2] The forecast for OA is alarming; with an ageing population, the prevalence is expected to rise dramatically in the coming decades. The direct healthcare costs of OA in various high-income countries account for 1%–2.5% of the gross domestic product.[3 4] OA can affect any joint and is most prevalent in the knee and hip, where it also leads to the greatest physical disability.[5] Due to a subsequent decrease in physical activity, hip OA also leads to more comorbidities and a higher age-adjusted mortality.[6]

Despite the tremendous burden of hip OA, there is no cure available. Therefore, current strategies focus on symptomatic treatment with only a modest effect.[7] This may partly result from a lack of knowledge on the aetiology, pathophysiology and risk factors of hip OA. Hip OA is a heterogeneous disease in which the risk factors and aetiology can differ widely from patient to patient. In contrast to knee OA, few large studies have focused on hip OA risk prediction so far. Up until 2022, 31 multivariable prediction models for incident knee OA have been published, while only 4 exist for hip OA. On top of that, all four have been created with data from Dutch cohort studies only.[8] This accentuates the need for more international collaborations in hip OA research.

Additionally, this lack of knowledge regarding person-specific risk factors for hip OA makes efficient and effective preventative and treatment strategies challenging, if not impossible, thus only one-size-fits-all treatment options for hip OA are available to date. Still, some risk factors for hip OA have been identified on a group level, such as age, gender,[5] obesity,[9] genetics,[10] race[11] and hip morphology (such as cam and pincer morphology or acetabular dysplasia).[12 13] However, each risk factor has only weak or even conflicting associations with hip OA, they have mainly been studied in single, heterogeneous cohorts, and are typically studied separately from each other. These single studies are underpowered to predict the risk of hip OA on an individual level.

Next to allowing for risk prediction on an individual level, a large dataset also allows for applying techniques from the rapidly emerging field of radiological image processing. These techniques could be used for classification of hip OA or diagnosing hip morphology. This does raise an additional question: how should we make optimal use of these techniques, both in a research setting and in clinical practice? Research on the use of artificial intelligence (AI) in hip OA, including the use of machine learning and deep learning, has so far been done in single cohort studies only.[14 15] This may limit the generalisability of the results and the continuation of the research into real-wold applications.

To overcome these challenges, we believe that the prospective cohort study design is ideal to better understand which individuals are at risk of developing hip OA and of progressing to end-stage disease. Harmonising data from multiple cohort studies into an individual participant-level database provides a large sample size, which may allow for individualised or at least subgroup-specific risk estimates. Further, large sample sizes and diverse cohorts from all over the world improve the generalisability of the findings.[16] To meet this need, the Worldwide Collaboration on OsteoArthritis prediCtion for the Hip (World COACH) consortium was initiated in 2018. The consortium aims to better understand risk factors for the development or progression of hip OA, and to optimise and automate methods for analysing radiological images of the hip. This will be pursued through studying multiple research questions within the consortium.

## CONSORTIUM DESCRIPTION
### Objectives and research questions

To study the research questions of the World COACH consortium, the consortium currently has five separate work packages. The first work package is Methodology, of which the goal is to discover, optimise, automate and validate new methods in OA research, such as an automated pipeline for hip morphological analyses and developing algorithms for the detection of radiographic hip OA (RHOA). This includes the application of AI. Hip Morphology is the second work package and it focuses on investigating associations between hip morphology and hip OA. Known morphological risk factors such as acetabular dysplasia, pincer morphology and cam morphology will be investigated, as well as general hip shape captured with statistical shape modelling. The third work package is Genetics, of which the aim is to study associations between genetics, hip morphology, environmental factors and OA, by applying Genome-Wide Associations Studies among other methods. The fourth work package is Clinical Measures, comprising physical examinations, questionnaires, quality of life and blood and urine samples. The aim is to study the associations between these measures and the development of hip OA. Finally, the fifth work package (Prediction Modelling) combines data and results from all other work packages to develop a personalised risk prediction model for the development and progression of hip OA, using both conventional and AI-driven methods.

### Cohort inclusion and consortium establishment

Prospective cohort studies were considered eligible if they had hip radiography—and optionally CT and/or MRI—available at two or more points in time, at least 4 years apart, and if they had a minimum of 200 participants at baseline. These criteria were applied at cohort level, but not participant level, thus having some participants with missing radiographs was not a reason to exclude a cohort. A systematic literature search was conducted in Embase, Ovid Medline and Cochrane CENTRAL to identify all studies that fulfilled the inclusion criteria. The search was first carried out in 2017 and was repeated in October 2020 and again in March 2023. Titles and abstracts were screened independently by two researchers (MMAvB and RA), and all described cohorts were further investigated, both by reading the full texts of the screened references and by additional internet searches. A Preferred

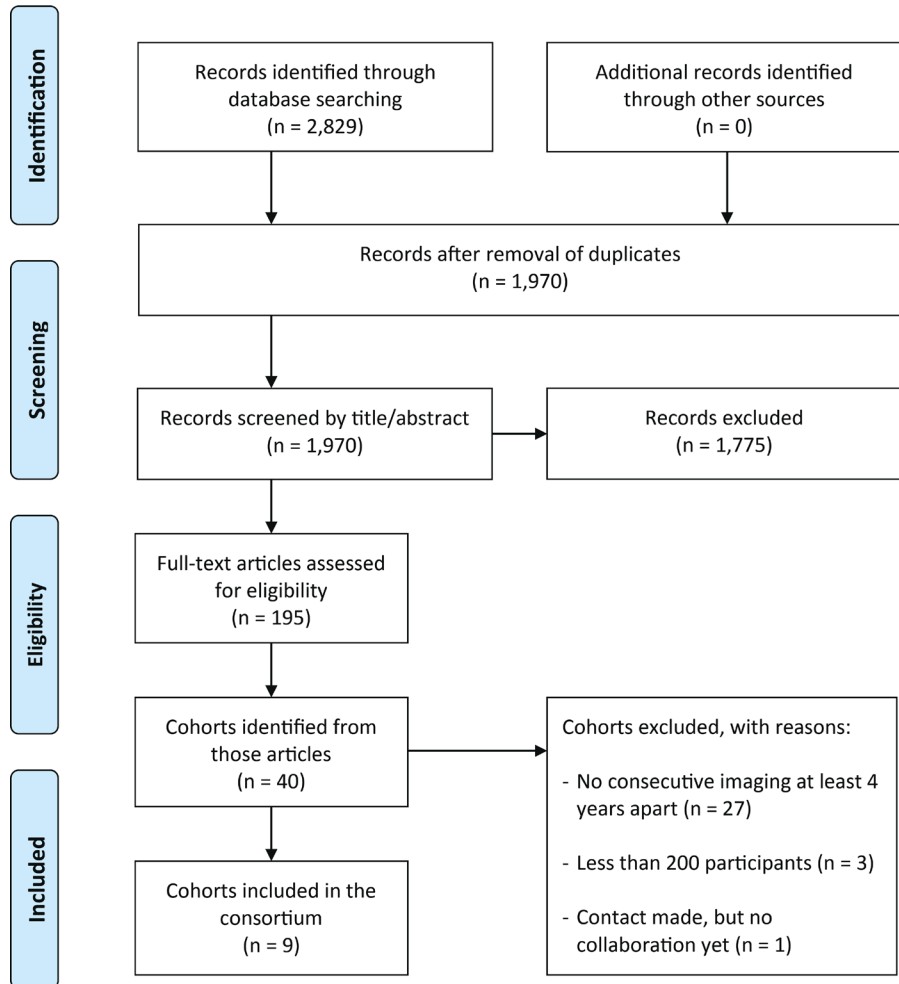

**Figure 1** PRISMA flow diagram detailing the literature search, screening and inclusion process for the World COACH consortium. PRISMA, Preferred Reporting Items for Systematic Reviews and Meta-Analyses; World COACH, Worldwide Collaboration on OsteoArthritis prediCtion for the Hip.

Reporting Items for Systematic Reviews and Meta-Analyses flow diagram of the search and inclusion process is presented in figure 1. In summary, we screened a total of 1970 records by title and abstract, of which we assessed 195 records in detail. We identified 40 study cohorts, 10 of which we considered eligible for the consortium. Investigators from the eligible cohorts were contacted and asked to collaborate. To date, 9 cohorts have been included in the consortium,[17–25] and contact has been initiated with the remaining eligible cohort.[26] The systematic search will be repeated every 2 years to identify newly eligible cohorts.

After the first search, the initiator of the World COACH consortium (RA) contacted the principal investigators of the nine identified cohorts to discuss the consortium's aims. A live meeting with principal investigators of eight cohorts, as well as other individuals interested in participating in the consortium, was held during the OsteoArthritis Research Society International (OARSI) world congress in Liverpool, 2018. During this meeting, the overall aims of the consortium were presented, and an inventory of the support for initiating the consortium was assessed. With unanimous support for the consortium, we decided to establish this initiative. Legal agreements for data sharing were drafted and executed. Since 2018, quarterly meetings have been held with the collaborators to determine the aims and research plans of the consortium.

### Description of the cohorts

A summary of the included cohorts can be found in table 1. All included studies are prospective cohort studies with a minimum of 5 and a maximum of more than 25 years of follow-up. Some cohorts still have ongoing data collection.[17 20 24] The earliest data collection in any cohort started in 1986,[18] and the most recent baseline data collection started in 2016.[20] Most included cohorts are population-based studies. Exceptions are the Cohort Hip and Cohort Knee (CHECK) cohort, in which participants had hip or knee complaints, Multicenter Osteoarthritis Study (MOST) and Osteoarthritis Initiative (OAI), which both included individuals with or at high risk of knee OA, and Femoroacetabular Impingement and Hip Osteoarthritis Cohort Study (FORCe), which included participants with hip and/or groin pain. At least six cohorts included non-white individuals,

**Table 1** Information and inclusion criteria of the included cohorts

| Cohort | Participants, inclusion criteria and recruitment | Age at enrolment | Races included | Follow-up time | Follow-up status |
|---|---|---|---|---|---|
| CHECK | Specific population: first visit to GP with hip/knee pain in the Netherlands Recruited through advertisements, newspaper articles, flyers, GP referral, friends and family | 45–65 years | W, B, A, O | 10 years | Completed |
| Chingford | General population: asymptomatic women Drawn from the register of a large general practice in Chingford, London, UK | 45–67 years | NS | 19 years | Completed |
| FORCe | Specific population: subelite soccer and Australian football players with hip and/or groin pain and a physical exam indicative of femoroacetabular impingement Recruited through advertisements, and from orthopaedic, sports medicine or physical therapy clinics | 18–50 years | NS | 5 years | Ongoing |
| JoCoOA | General population: black and white men and women in a rural area Drawn by probability sampling from the population of Johnston County, North Carolina, USA | ≥45 years | W, B | 25+years | Completed |
| MOST | Specific population: individuals with existing knee OA or those at high risk for it Identified through health insurance companies, voter registration tapes, commercial list brokers and other sources Recruited by mailings of letters and study brochures to eligible individuals | 50–79 years | W, B, A, O, NS | 7 years | Ongoing |
| OAI | Specific population: individuals with existing knee OA or those at high risk for it Recruited through focused mailings, advertisements in local newspapers, presentations at church, community or civic meetings, and a website about knee pain and osteoarthritis | 45–79 years | W, B, A, O, NS | 8 years | Completed |

van Buuren MMA, *et al. BMJ Open* 2024;**14**:e077907. doi:10.1136/bmjopen-2023-077907

**Table 1** Continued

| Cohort | Participants, inclusion criteria and recruitment | Age at enrolment | Races included | Follow-up time | Follow-up status |
|---|---|---|---|---|---|
| RS | General population: people living in the Ommoord district in Rotterdam, the Netherlands<br>Drawn from the municipal register, after which random clusters of potential participants got invited through a letter sent to their home | ≥ 45 years | W, B, A, Mix | 25 years | Ongoing |
| SOF | General population: community-dwelling women<br>Identified through health insurance companies, jury selection lists, voter registration lists and drivers' licenses and identification cards lists<br>Recruited by mailings of letters and study brochures to eligible individuals | ≥ 65 years | W, B | 8 years | Completed |
| TASOAC | General population: community-dwelling men and women<br>Randomly selected from the electoral roll in Southern Tasmania | 50–80 years | W, NS | 10 years | Completed |
| World COACH | Mix of individuals from the general population, those with possible early hip or knee OA, those with pre-existent knee OA or at high risk for it, and subelite football players with hip/groin pain and possible femoroacetabular impingement | 18–80 years | W, B, A, O, Mix | 5–25 years | Ongoing |

Race abbreviations: W, white; B, black; A, Asian; O, other; NS, not specified; Mix, admixture.
CHECK, Cohort Hip and Cohort Knee; Chingford, Chingford 1000 women study; FORCe, Femoroacetabular Impingement and Hip Osteoarthritis Cohort Study; GP, general practitioner; JoCoOA, Johnston County Osteoarthritis Project; MOST, Multicenter Osteoarthritis Study; OA, osteoarthritis; OAI, Osteoarthritis Initiative; RS, The Rotterdam Study; SOF, Study of Osteoporotic Fractures; TASOAC, Tasmanian Older Adult Cohort; World COACH, Worldwide Collaboration on OsteoArthritis prediCtion for the Hip.

and six cohorts studied both men and women. The total World COACH population includes participants from three continents. Across cohorts, participants were aged 18–80 years at enrolment. The aims and methods of each individual cohort are shortly described, emphasising those characteristics that correspond with the consortium's inclusion criteria (eg, radiographic protocols are highlighted, but not CT or MRI).

### Cohort Hip and Cohort Knee study

The CHECK study was a multicentre prospective cohort study in the Netherlands that ran from 2002 until 2015 for a total of 10 years follow-up.[25] The aim was to study the course, prognosis and underlying mechanisms of early symptomatic OA. The study included 1002 participants aged 45–65 years, with a first episode of pain in the hip and/or knee. Participants were eligible if they had not yet visited a general practitioner (GP) or were within 6 months of their first visit to the GP for these symptoms, or if they had never visited a GP before for these symptoms, and if there was no other diagnosis that could explain the symptoms at the time of inclusion. Participants were recruited between October 2002 and September 2005, mostly through local newspaper articles and advertisements, and

the website of the Dutch Arthritis Society (69% of inclusions). Additionally, eligible individuals were referred by their GP to 1 of 10 participating general and university hospitals (6%), recruited through a flyer, family member or a friend (12%), and for the remainder it was not recorded. Standardised weight-bearing anteroposterior (AP) hip or pelvic radiographs, using a wedge to get the hips in 15° internal rotation, were obtained at baseline and at 2, 5, 8 and 10 years of follow-up. At 10 years follow-up, 87% of the baseline cohort had RHOA scores completed.

### The Chingford 1000 women study

The Chingford study was a population-based prospective cohort study that aimed to assess musculoskeletal disease in the female population.[19] It ran from 1989 to 2010, having over 20 years of follow-up. The study recruited asymptomatic female participants, aged 45–64 years, from the registry of a large general practice (over 11 000 patients) in Chingford, London, UK. All 1353 women in that age range were invited to participate, of which 1003 were included. Standardised supine AP pelvic radiographs, using a small sand bag under the knees to minimise hip rotation, were obtained at years 2, 8 and 20 of follow-up.[27] After 8 years, 99% of the participants who had baseline RHOA scores, also had a follow-up score. At 15 years, 77% of the original cohort were still being followed up.

### Femoroacetabular impingement and hip OsteoaRthritis Cohort study

The FORCe study is an ongoing prospective cohort study aiming to evaluate changes in hip joint structure in subelite soccer and Australian football players with hip and/or groin pain, with a focus on early hip OA features.[17] Participants were recruited between August 2015 and October 2018. The study included 239 participants, aged 18–50 years, who were recruited through advertisements at sporting venues and from orthopaedic, sports medicine or physical therapy clinics. Participants were eligible if they had self-reported hip and/or groin pain for >6 months, with a gradual onset, and pain with a score between 3 and 8 on an 11-point numerical rating scale. They also had to have a positive flexion-adduction-internal-rotation test in at least one hip, indicative of femoroacetabular impingement. At baseline, all participants underwent standardised supine AP pelvic radiographs with the feet in 15° internal rotation using a positioning aid, and MRI of the hips. The study is currently inviting participants for a 5-year follow-up visit, that comprises pelvic radiography and MRI according to the same standardised protocols.

### Johnston County Osteoarthritis Project (JoCoOA)

The JoCoOA was a population-based cohort study with up to 30 years of follow-up.[23 28] Its aim has been to examine the incidence, prevalence and progression of OA in black and white men and women in a rural county. The study started in 1991 and data collection ended in 2018. Participants, all non-institutionalised black and white men and women, were drawn by probability sampling from the population of Johnston County, North Carolina, USA. The study included 4337 participants aged ≥45 years. Standardised supine AP pelvic radiographs with the feet in 15° internal rotation were obtained at baseline, and then every 5–6 years, except for women under the age of 50 at the time of assessment (per protocol). Follow-up rates over the years have been between 50% and 60% for each subsequent visit, with the main reason for loss to follow-up being death (around 17% each visit).

### Multicenter Osteoarthritis Study

MOST is a multicentre prospective cohort study in the USA that started in 2003 and has followed participants for 20 years so far.[24] The aim was to study risk factors for the development and progression of knee OA and knee pain. Two centres in Birmingham (Alabama) and Iowa City (Iowa) recruited participants with pre-existing knee OA or those at high risk for knee OA from the general population. Eligible individuals were identified through databases from health insurance companies, voter registration tapes, commercial list brokers and other sources, after which they were sent invitation letters and study brochures. The study included 3026 individuals aged 50–79 years in its initial phase, with a new cohort of 1500 individuals included in 2016–2018. A standardised weight-bearing AP full-limb radiograph of the lower extremities (including the pelvis) with the tibial tubercles facing forward and the X-ray beam centred at the knee was obtained at baseline, and again at 5 years of follow-up. Because the pelvis was included in these sequential full-limb radiographs, this cohort study on knee OA was also eligible for inclusion in the consortium. After 5 years, 99% of participants that had baseline RHOA scores also had follow-up scores completed.

### Osteoarthritis Initiative

The OAI study was a multicentre prospective cohort study of knee OA in the USA.[22 29] OAI aimed to provide resources to enable a better understanding of prevention and treatment of knee OA. It was initiated in 2002 and the entire cohort finished its 8-year follow-up in 2015, but the follow-up continues for certain subsets of participants. The OAI study has included 4796 participants with pre-existing knee OA or those at high risk for developing knee OA, from the general populations of Baltimore (Maryland), Columbus (Ohio), Pittsburgh (Pennsylvania), and Pawtucket (Rhode Island). Participants were aged 45–79 years at enrolment. These participants were contacted through focused mailings, advertisements in local newspapers, presentations at church, community or civic meetings, and a website about knee pain and OA. Standardised weight-bearing AP pelvic radiographs using a v-shaped foot-positioning frame to get the feet in 5° of internal rotation were obtained at baseline, 4 years and 8 years follow-up. The inclusion of pelvic radiography

made this knee OA study also eligible for the consortium. RHOA scores were available for 77% at the 4-year follow-up visit, while the 8-year radiographs have yet to be scored for hip OA.

### The Rotterdam Study (RS)

The RS is an ongoing prospective population-based cohort study in a district of the city of Rotterdam, the Netherlands.[30] It aims to address determinants and occurrence of cardiovascular, neurological, musculoskeletal, ophthalmologic, psychiatric and endocrine diseases in the elderly. After the pilot in 1989, the study started recruiting in 1990, and it currently has over 25 years of follow-up. The names and addresses of eligible participants were drawn from the municipal register, after which random clusters of potential participants got invited through a letter sent to their home, followed up by a phone call. Up to 2008, the study had included 14926 participants (72% of 20744 invitees) aged ≥45 years, divided into 3 subcohorts from different enrolment periods, namely RS-I, RS-II and RS-III. Recruitment of a fourth subcohort (RS-IV) started in 2016 and has recently been finished. Data from RS-IV will also be included once they fulfil the inclusion criteria. Standardised weight-bearing AP pelvic radiographs with the feet in 10° internal rotation were obtained at baseline, and then approximately every 4–6 years. Because of the different subcohorts with different follow-up schemes, there is no single follow-up rate. The follow-up rate decreases over time, especially after 12 years and over, as can be expected in an ageing population.

### Study of Osteoporotic Fractures (SOF)

SOF was a multicentre prospective population-based cohort study of community-dwelling women aged ≥65 years.[18] The primary purpose of SOF was to describe risk factors for osteoporotic fractures. Women were recruited between September 1986 and October 1988 from 4 metropolitan areas in the USA: Baltimore (Maryland), Pittsburgh (Pennsylvania), Minneapolis (Minnesota) and Portland (Oregon). Eligible women were identified in multiple ways: through membership lists from health insurance companies, jury selection lists, voter registration lists and drivers' licenses and identification cards lists. Women received a letter and brochure inviting them to participate. The original cohort included 9704 mostly Caucasian women who had not undergone bilateral hip replacement and were able to walk without assistance. The cohort has over 20 years of prospective data about osteoporosis. Standardised supine AP pelvic radiographs with the hips in 15–30° internal rotation were obtained at baseline and after 8 years of follow-up. The follow-up rate for RHOA scores was 100%.

### Tasmanian Older Adults Cohort (TASOAC) study

The TASOAC study is an ongoing prospective population-based cohort study of 1099 community-dwelling men and women, aged 50–80 years.[21] The study aimed to identify factors associated with the development and progression of OA in multiple joints, including the hip. Eligible participants were randomly selected from the electoral roll in Southern Tasmania, using sex-stratified simple random sampling without replacement (response rate 57%). Participants were excluded if they were institutionalised or if they reported a contraindication for MRI. Enrolment started in 2002 and the cohort had follow-up moments at approximately 2.7 years, 5 years and 10 years. Standardised weight-bearing AP pelvic radiographs with the feet in 10° internal rotation were obtained at baseline and after 10 years of follow-up. A subgroup (n=250) had MRI of the right hip in the sagittal plane at 2.7 and 5 years follow-up.[31] At inclusion, the TASOAC study did not yet have OA scores available for their 10-year follow-up. These will be added at a later time point.

## DATA HARMONISATION

Retrospective harmonisation is an intricate process, considering few original studies have used identical collection methods and procedures. Our harmonisation process will be based both on expert opinion within the consortium, as well as on the Maelstrom Research guidelines for rigorous retrospective data harmonisation.[32]

### Defining the DataSchema

We started by analysing the present literature on the included studies (eg, study protocols, published papers) to evaluate sources of study heterogeneity. The next step was to define variables and evaluate the harmonisation potential. All available variables from individual studies within the consortium were identified and systematically entered in a DataSchema[32], categorised in 13 sections: demographic data, physical examinations and anthropometry, radiographic measurements of OA, questionnaires, family history, procedures, biospecimens, lifestyle and diet, comorbidities, medication, physical and cognitive functioning, quality of life and genetics. This allowed us to evaluate comparability between studies. Next, all data were catalogued based on their characteristics. All similar variables that indicate the same measurement were grouped together and renamed using a common pooled variable. Finally, the process of data harmonisation was initiated, for which we used and will continue to use one of the established approaches, depending on the data[32]:

► Simple calibration model: will be used to transform continuous variables into new continuous variables (eg, transforming height in inches to height in centimetres). The distribution of the values will be compared across cohorts to assess for differences within the measurement.
► Algorithmic transformation: will be used to harmonise continuous or categorical variables with combinable ranges or categories (eg, race or ethnicity, education level).
► Standardisation model: will be used to harmonise the same constructs measured with different scales, when

there are no bridging items available (eg, two independent questionnaires on hip symptoms)
► Latent variable model: will be used to harmonise variables with different scales that have some bridging items (eg, OA grade based on the Kellgren-Lawrence (KL), Croft or OARSI atlas classification, which all contain items such as joint space narrowing and osteophytes).

### Data storage and processing

After establishing data transfer agreements with each included cohort, all required individual participant data were transferred to a central server. Variables were then prepared to be entered into a relational database using the Observational Medical Outcomes Partnership (OMOP) Common Data Model (CDM) structure.[33 34] The CDM is a 'person-centric' model and is optimised for observational research purposes such as identifying patient populations with certain outcomes (such as hip OA), characterisation of these populations for various parameters (including risk factors), and predicting the occurrence of the outcome in individuals. Variables were clustered in domains using the CDM's standardised vocabularies. Although the variables were mapped to the standardised vocabularies, we also stored the original source values, to ensure that all data entries can be traced when locating or preventing unforeseen errors. The OMOP CDM does not require specific software and can be realised in any relational database software. We currently use an advanced open-source relational database system (PostgreSQL V.15.2, PostgreSQL Global Development Group) which uses the SQL data definition and query language. The stored variables from each cohort and their individual participants include demographics and follow-up visits, along with measurements and procedures performed at each visit. This original data collection setup is used within the relational database model, which contains seven linked tables (figure 2). The person table contains demographics of the included individual, such as biological sex, year of

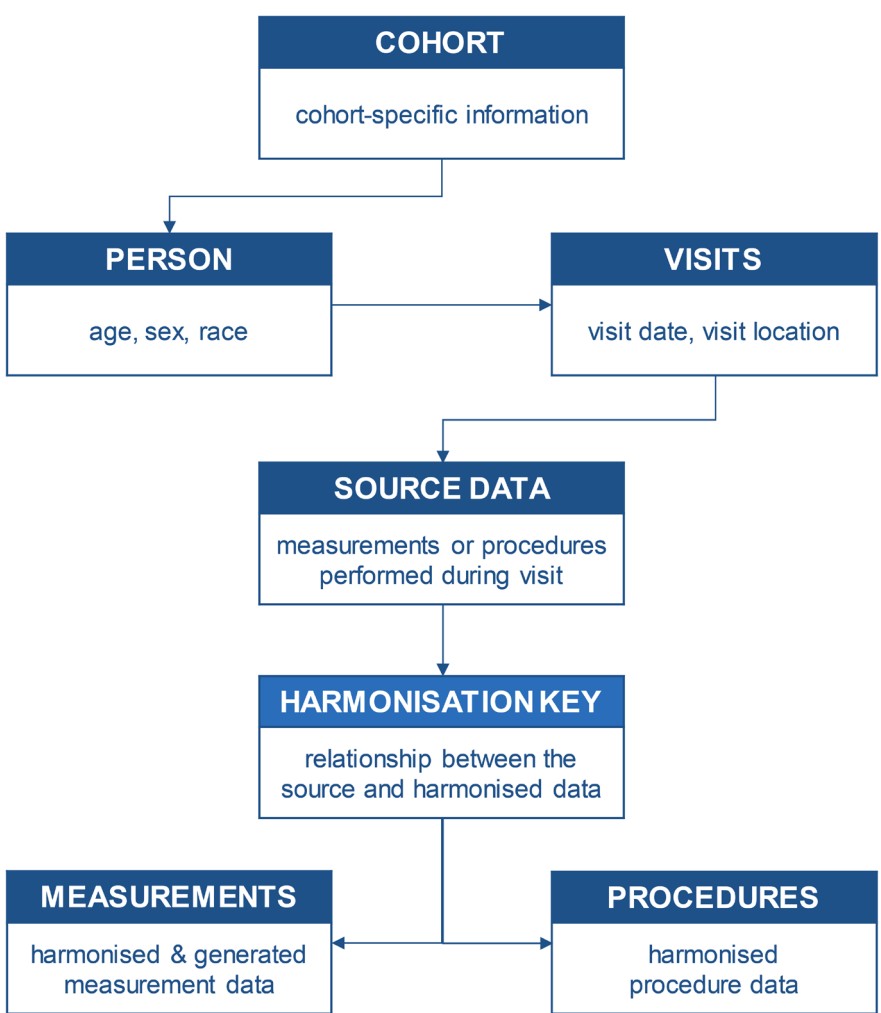

**Figure 2** A simplified schema of the relational database used in the World COACH consortium. World COACH, Worldwide Collaboration on OsteoArthritis prediCtion for the Hip.

birth and the originating cohort identification number, which links to the descriptive cohort table. The visit occurrence table contains the different time points and study sites (where available) at which data were collected from each individual. The measurement and procedure occurrence tables contain harmonised and newly generated World COACH variables for each specific follow-up visit (eg, harmonised RHOA score based on available KL or modified Croft grades). The harmonisation steps are documented and harmonised values are linked to their source value through the harmonisation key table.

## Outcome measures

The main outcome is the development of hip OA within the various follow-up periods, although there are several outcomes of interest for secondary analyses. Hip OA could be defined structurally by radiological indices, clinically by pain and/or functional indices, and if possible, by a combination of these two. Radiographs are the only validated and recommended imaging modality to investigate hip OA as a structural outcome.[35 36] Pelvic radiographs have been read for the presence and severity of radiographic OA using either the KL classification,[37 38] the (modified) Croft classification,[39–41] or the atlas of individual radiographic features in osteoarthritis (OARSI atlas).[42] The inclusion criterion of having hip radiography available at two or more points in time, at least 4 years apart, was set to determine the presence or absence of RHOA at both time points. This is necessary to distinguish between incident RHOA (in case of no RHOA at baseline) or progression of RHOA (in case of RHOA at baseline). Different pain scores, such as visual analogue scales, Western Ontario and McMaster Universities Osteoarthritis Index pain scale, the Hip Disability and Osteoarthritis Outcome Score and other scores will be harmonised into a single pain index if and where possible.

## FINDINGS TO DATE

As of submission, the World COACH database contains data on 39805 individuals. The mean age of all World COACH participants at baseline was 65.4±8.8 years and the study sample consisted of 27957 (70.2%) women. Mean body mass index (BMI) ranged from 24.6 kg/m$^2$ in the FORCe study to 30.7 kg/m$^2$ in MOST, with an overall mean BMI of 27.5 kg/m$^2$. The amount of available baseline pelvic and/or hip radiographs is 34257 (table 2).

The amount of currently available RHOA scores in the included studies is shown in online supplemental table S1, both at baseline and at each of the studies' respective follow-up visits. At baseline, there were almost 60000 hips with a valid RHOA score available. At both the 4-year and the 5-year mark, 10000 hips were scored, and around 20000 hips have a score after 8 years of follow-up. The numbers of hips with a valid OA score logically decrease with longer follow-up times (10, 12, 15, 20 and 25 years of follow-up). The number of RHOA scores available to date may increase in future publications as we plan to score available radiographs that currently miss radiographic OA scores.

Currently available baseline RHOA scores are shown in table 3. Most included cohorts used several methods for RHOA scoring such as KL, modified Croft and OARSI individual features. At baseline, 36065 hips (61.1% of those with available RHOA scores) showed no signs of RHOA (score 0), and 17778 hips (30.1%) had early or doubtful RHOA (score 1). Definite RHOA or a total hip replacement was present in 5135

**Table 2** Baseline characteristics of the included cohorts

| Cohort | N baseline participants | N participants with data available | N baseline pelvic/hip radiographs | Mean (SD) age, years | Mean (SD) height, cm | Mean (SD) weight, kg | Mean (SD) BMI, kg/m$^2$ | % Female |
|---|---|---|---|---|---|---|---|---|
| CHECK | 1002 | 1002 | 1002 | 55.9 (5.2) | 170 (8) | 75.5 (13.8) | 26.2 (4.0) | 79 |
| Chingford | 1003 | 1003 | 1003 | 54.2 (6.0) | 162 (6) | 66.9 (11.8) | 25.6 (4.3) | 100 |
| FORCe | 239 | 239 | 239 | 27.2 (5.9) | 178 (9) | 78.5 (12.8) | 24.6 (3.1) | 22% |
| JoCoOA | 4337 | 4010 | 4004 | 62.2 (10.0) | 166 (10) | 81.4 (18.3) | 29.5 (6.2) | 62 |
| MOST | 3026 | 3026 | 3008 | 62.5 (8.1) | 169 (12) | 87.9 (18.9) | 30.7 (6.0) | 60 |
| OAI | 4796 | 4796 | 4771 | 61.2 (9.2) | 165 (24) | 81.2 (16.4) | 28.6 (4.8) | 58 |
| RS | 14926 | 14926 | 11147 | 66.0 (10.5) | 168 (10) | 75.9 (13.9) | 26.9 (4.1) | 59 |
| SOF | 10366 | 9704 | 8291 | 71.6 (5.2) | 159 (6) | 67.0 (12.0) | 26.4 (4.5) | 100 |
| TASOAC | 1099 | 1099 | 1099 | 63.0 (7.5) | 167 (9) | 77.9 (15.0) | 27.9 (4.8) | 51 |
| World COACH | 40794 | 39805 | 34257 | 65.4 (8.8) | 166 (12) | 75.7 (14.7) | 27.5 (4.7) | 70 |

BMI, body mass index; CHECK, Cohort Hip and Cohort Knee; Chingford, Chingford 1000 women study; FORCe, Femoroacetabular Impingement and Hip Osteoarthritis Cohort Study; JoCoOA, Johnston County Osteoarthritis Project; MOST, Multicenter Osteoarthritis Study; OAI, Osteoarthritis Initiative; RS, The Rotterdam Study; SOF, Study of Osteoporotic Fractures; TASOAC, Tasmanian Older Adult Cohort; World COACH, Worldwide Collaboration on OsteoArthritis prediCtion for the Hip.

**Table 3** Indication of radiographic hip osteoarthritis scores at baseline

| Cohort | N total hips | Available RHOA scores | 0 | 1 | 2 | 3 | 4 | THR | N missing |
|---|---|---|---|---|---|---|---|---|---|
| CHECK | 2004 | KL (OARSI) | 1225 | 458 | 204 | 13 | 0 | 0 | 104 |
| Chingford | 2006 | KL (OARSI) | 1031 | 78 | 152 | 5 | 1 | 1 | 738 |
| FORCe | 465 | KL | 446 | 19 | 0 | 0 | 0 | 0 | 0 |
| JoCoOA | 8020 | KL (OARSI) | 1837 | 4286 | 1493 | 91 | 43 | 0 | 270 |
| MOST | 6052 | KL (OARSI) | 2194 | 1004 | 354 | 75 | 7 | 0 | 2418 |
| OAI | 9592 | OA score (based on modified Croft), OARSI | 7121 | 1064 | 543 | NA | NA | NA | 864 |
| RS1 | 16240 | KL (OARSI) | 6406 | 4110 | 426 | 63 | 18 | 216 | 5001 |
| RS2 | 6024 | KL (OARSI) | 3584 | 669 | 105 | 15 | 5 | 62 | 1584 |
| RS3 | 7878 | KL (OARSI) | 5282 | 532 | 58 | 4 | 2 | 42 | 1958 |
| SOF | 19414 | Modified Croft (OARSI) | 5725 | 5558 | 262 | 117 | 10 | 0 | 7742 |
| TASOAC | 1962 | OA score (based on OARSI), OARSI | 1214 | NA | 748 | NA | NA | NA | 0 |
| World COACH | 79657 | Mix | 36065 | 17778 | 4345 | 383 | 86 | 321 | 20679 |

Scores 0–4 are either KL, modified Croft, or a custom summary OA score. OARSI scores are not shown in this table.
CHECK, Cohort Hip and Cohort Knee; Chingford, Chingford 1000 women study; FORCe, Femoroacetabular Impingement and Hip Osteoarthritis Cohort Study; JoCoOA, Johnston County Osteoarthritis Project; KL, Kellgren & Lawrence; MOST, Multicenter Osteoarthritis Study; NA, not applicable; OAI, Osteoarthritis Initiative; OARSI, Osteoarthritis Research Society International individual features scores; RHOA, radiographic hip osteoarthritis; RS, The Rotterdam Study; SOF, Study of Osteoporotic Fractures; TASOAC, Tasmanian Older Adult Cohort; THR, total hip replacement; World COACH, Worldwide Collaboration on OsteoArthritis prediCtion for the Hip.

hips at baseline (8.7%). When looking at hips with both baseline and follow-up RHOA scores available, 42619 hips were free of definite RHOA at baseline. Within this group, 3207 (8%) of the hips developed incident RHOA at follow-up (online supplemental table S2).

Other available variables of which the harmonisation process is still ongoing (besides those shown in the tables) are ethnicity/race (all cohorts), socioeconomic status (all cohorts), smoking status (all cohorts), hip pain (CHECK, Chingford, FORCe, JoCoOA, MOST, RS, TASOAC), hip range of motion (CHECK, FORCe, JoCoOA), bone mineral density (Chingford, OAI, SOF, RS, TASOAC) and physical activity (CHECK, Chingford, FORCe, JoCoOA, MOST, OAI, RS, TASOAC).

## STRENGTHS AND LIMITATIONS
The main strength of the World COACH consortium is its rich variety of harmonised, individual participant data from all available prospective cohort studies on hip OA worldwide. Although this offers significant challenges, it has the potential to improve the generalisability of our findings. The large sample size offers unique opportunities to study the relationship between different risk factors and the development and progression of hip OA on an individual level, as well as the identification of high-risk subgroups. It also allows for analysis of interactions between these factors, such as the effect of obesity across different hip shape variations. This will hopefully allow for the creation of the first person-specific and/or subgroup-specific risk estimation of developing hip OA. This personalised model can in turn be used to identify

both high-risk individuals and the factors that contribute to this risk. In turn, this provides opportunities for future studies on prevention and individualised OA treatment. Furthermore, the World COACH consortium strives to offer solutions to some of the greatest epidemiological issues in terms of hip OA research by testing, automating and validating methodological issues related to image analysis, with the potential of providing a benchmark for imaging analysis in hip OA research. Finally, the extensive dataset allows for investigating an array of secondary research questions along with the main aims of the consortium.

The strength of a relational database is that it is possible to enrich the existing consortium data with data from new cohorts once they meet the inclusion criteria, without the need to restructure the datasets. The flexible structure of relational databases allows for seamless expansion to handle increasing volumes of data and it can easily adapt to frequent updates or deletions.

Limitations of the consortium include the limited geographic selection of cohorts from the Western world (Australia, Europe and the United States). To date, no cohorts have been included from Africa, Asia or South America, which may limit the generalisability of findings. There is some heterogeneity in the populations from which World COACH participants were originally drawn. Most cohorts have included participants from the general population, although with an age restriction (Chingford, JoCo, the Rotterdam Study, SOF and TASOAC), but some cohorts included participants with specific characteristics (CHECK, FORCe, MOST, OAI). This may limit generalisability of the findings and is something we have to account

van Buuren MMA, *et al. BMJ Open* 2024;**14**:e077907. doi:10.1136/bmjopen-2023-077907

for in future analyses. We will consider the use of different statistical methods that could account for cohort differences and address heterogeneity. Most cohorts included participants aged 45 years or older, while only the FORCe cohort included younger participants. Although people aged over 45 years represent the vast majority of the hip OA population, we will be underpowered to externally validate findings in people younger than 45 years. The World COACH consortium is also limited by the heterogeneity in collection of variables by the cohorts, which was inherently done in slightly different ways. This requires harmonisation of variables, which is mainly based on (potentially subjective) expert-based criteria. On the other hand, pooling of the data creates far greater statistical power than previously possible. Finally, although a subset of the data consists of 3-dimensional imaging data such as CT or MRI, most analyses will be performed using plain AP pelvic radiographs. As stated by the American College of Rheumatologists (USA),[43] the National Institute for Health and Care Excellence (UK)[44] and the European Alliance of Associations for Rheumatology (EU),[45] imaging is not necessary for the diagnosis of hip OA in clinical practice. Still, radiographs probably contain valuable (hidden) predictive information for hip OA, and they are extensively used in daily clinical practice. Radiographs are also the only valid method to diagnose structural hip OA so far and are a simple and inexpensive tool for use in large clinical studies. Findings from this consortium may also guide primary care providers as to which patients should be sent for radiographic imaging, and which patients could start conservative treatment based on a clinical diagnosis of hip OA.

## COLLABORATION

We will provide a harmonised database containing all prospective data on hip OA. We encourage the use of data by third parties, although this is subject to approval by the steering committees of the World COACH consortium and the participating cohorts, as well as to legal boundaries regarding data ownership. To streamline the processing of third-party requests, we have developed a standardised data request form that can be distributed and reviewed uniformly. This will ensure consistency in the way data requests are handled within World COACH.

Our approach to data storage involves the Findable, Accessible, Interoperable and Reusable (FAIR) principles.[46] This will be achieved by using unique and persistent identifiers, by adhering to the Observational Health Data Sciences and Informatics terminology where possible, and by implementing standardised access protocols to make data available on request. By adhering to the FAIR principles, we aim to promote collaboration and transparency to advance scientific research in the field of hip OA and beyond. The relational database supports data storage that is compatible with other data sources and formats, enabling seamless integration.

Finally, the project will be overseen by two committees: a steering committee and an advisory committee, which have quarterly meetings regarding the consortium. Both committees consist of a diverse team of experienced researchers and clinicians in the areas of OA, rheumatology, epidemiology and image processing. Their combined expertise will provide valuable guidance and ensure the project's success. More information on the consortium and on data requests can be obtained from the website: www.worldcoachconsortium.com.

## PATIENT AND PUBLIC INVOLVEMENT

A patient and public committee (PPC) is being formed to ensure that the wider public is represented in World COACH. World COACH aims to ensure that all projects are relevant, meaningful and have impact on the people and patients it aims to serve. This includes not only patients with hip OA, but also families, caregivers and members of the general public. The PPC will be involved in prioritising research questions and in helping to shape the long-term vision of World COACH with particular consideration for the interests of the public and patients with hip OA. Our goal is to engage with a relevant population by promoting our project at various events. Additionally, we have made our research team accessible to the public through the World COACH website, where individuals can contact us directly via email, and through public meetings such as a local 'OA cafe'. We actively encourage such interactions in our presentations to foster engagement and promote greater understanding of our research to the public, and for the team to better understand what is relevant and important to patients.

**Author affiliations**
[1]Department of Orthopaedics and Sports Medicine, Erasmus Medical Center, Rotterdam, Zuid-Holland, Netherlands
[2]Institute for Medical Research, University of Tasmania Menzies, Hobart, Tasmania, Australia
[3]Department of Orthopedics, UMC Utrecht, Utrecht, Netherlands
[4]Orthopaedic-Biomechanics Research Group, Department of Mechanical Engineering, Faculty of Engineering, University of Birjand, Birjand, Iran
[5]Department of Orthopaedics Rheumatology and Musculoskeletal Sciences, University of Oxford Nuffield, Oxford, UK
[6]Department of General Practice, Erasmus Medical Center, Rotterdam, Zuid-Holland, Netherlands
[7]Department of Internal Medicine, Erasmus Medical Center, Rotterdam, Zuid-Holland, Netherlands
[8]Department of Epidemiology and Preventative Medicine, Monash University, Melbourne, Victoria, Australia
[9]Centre for Imaging Sciences, The University of Manchester, Manchester, UK
[10]La Trobe Sport and Exercise Medicine Research Centre, La Trobe University School of Allied Health Human Services and Sport, Melbourne, Victoria, Australia
[11]Boston University School of Medicine, Boston, Massachusetts, USA
[12]Department of Radiology, UMC Utrecht, Utrecht, Netherlands
[13]Department of Medicine, University of California Davis School of Medicine, Sacramento, California, USA
[14]Department of Epidemiology and Biostatistics, University of California San Francisco, San Francisco, California, USA
[15]Thurston Arthritis Research Center, The University of North Carolina at Chapel Hill, Chapel Hill, North Carolina, USA
[16]Department of Radiology & Nuclear Medicine, Erasmus Medical Center, Rotterdam, Zuid-Holland, Netherlands

[17]Department of Biomechanical Engineering, TU Delft, Delft, Zuid-Holland, Netherlands

**Acknowledgements** We would like to thank all participants of the cohort studies that are involved in the World COACH consortium. We gratefully acknowledge all international organisations that collaborated with the cohort studies in World COACH, as well as the Osteoarthritis Research Society International (OARSI) for endorsing the World COACH consortium. We thank the (non-profit) funding bodies who financially support the World COACH consortium: the Dutch Arthritis Society (grant no. 18-2-203 and 21-1-205), the Dutch Research Council (NWO Veni grant scheme no. 09150161910071) and the Erasmus MC, University Medical Center, Rotterdam (Erasmus MC Fellowship). For the purposes of open access, the authors have applied a CC BY public copyright licence to any Author Accepted Manuscript version arising from this submission

**Contributors** RA initiated the study. RA, MMAvB, NSR, MAvdB, FDEMB, NKA, SMAB-Z, CGB, FC, TFC, DF, WPG, GJ, SK, NEL, CL, JVM, AEN, MN, JR and HW worked on the conceptual design of the study. MMAvB and RA identified eligible cohorts and contacted cohort investigators for collaboration. MMAvB, RA, NSR, MAvdB, HA, KC, JH, SK, JAL, JVM, ABM, AEN, MN, JT and HW collected the existing cohort data. MMAvB, NSR, MAvdB, FDEMB, JT and RA have worked on the database and on the harmonisation process. MMAvB, NSR, MAvdB, FDEMB, VA, CGB, TFC, WPG, CL, JAL, JVM, AEN, MN, EO, JR, JT and RA have worked on (preliminary) statistical analyses so far. All authors critically reviewed and revised the manuscript and contributed to interpretation of the data. All authors read and approved the final version of the manuscript. MMAvB acts as guarantor and accepts full responsibility for the finished work and/or the conduct of the study, had access to the data, and controlled the decision to publish.

**Funding** The World COACH consortium itself has been funded through research grants by the Dutch Arthritis Society (grant no. 18-2-203 and 21-1-205), the Dutch Research Council (NWO Veni grant scheme no. 09150161910071), and Erasmus MC, University Medical Center Rotterdam (Erasmus MC Fellowship). MvB is funded by the Dutch Arthritis Society (research grants 18-2-203 and 21-1-205) and by an Erasmus MC Fellowship grant. NR, MvdB and FB are funded by the Dutch Arthritis Society (research grant 21-1-205). NA is funded by the Versus Arthritis Centre for Sport, Exercise & Osteoarthritis Research. SBZ is funded through independent research grants by the European Research Council (ERC), the Dutch Arthritis Society, and The Netherlands Organisation for Health Research and Development (ZonMw). CB and JvM are funded by the Dutch Arthritis Society (LLP-34). TC is funded by research grants from the Engineering and Physical Sciences Research Council (EPSRC) UK, the Medical Research Council (MRC) UK, and the Wellcome Trust. KC and JH are funded by the National Health and Medical Research Council (NHMRC) Australia (GNT GNT1088683), AM is funded by an NHMRC Australia Early Career Fellowship (GNT1156674). DF is funded by a research grant from the National Institutes of Health (NIH) (AR072571). CL is funded by a research grant from the MRC UK (MR/S00405X/1) as well as a Sir Henry Dale Fellowship jointly funded by the Wellcome Trust and the Royal Society (223267/Z/21/Z). This research was funded in whole, or in part, by the Wellcome Trust (Grant number 223267/Z/21/Z). AN is funded by the Centers for Disease Control and Prevention (CDC) (U01DP006266 and U01DP003206; Association of Schools of Public Health/CDC S043, S1734, S3486), and the NIH and National Institute of Arthritis and Musculoskeletal and Skin Diseases (NIAMS) (P60AR30701, P60AR049465, P60AR064166 and P30AR072580). MN is funded by research grants from the NIH. JT is funded by the China Scholarship Council (CSC). HW is funded by research grants from Interreg, Kansen voor West, NWO, the Innovative Medicines Initiative (IMI), and the Dutch government. RA is funded by the Dutch Arthritis Society (research grants 18-2-203 and 21-1-205), the Dutch Research Council (NWO Veni grant scheme no. 09150161910071), and Erasmus MC, University Medical Center Rotterdam (Erasmus MC Fellowship).

**Competing interests** GJ reports personal fees from Novartis outside the submitted work. SBZ reports consulting fees from Pfizer Infirst Healthcare and personal fees for being a Deputy Editor for Osteoarthritis and Cartilage outside the submitted work. CL and TC report a patent for an image processing apparatus and method for fitting a deformable shape model to an image using random forest regression voting. CL reports licensing royalties for this patent from Optasia Medical outside the submitted work. AN is an associate editor for Osteoarthritis and Cartilage and is on the OARSI Board of Directors outside the submitted work. AM is on the Editorial Board for the British Journal of Sports Medicine and the Journal of Science and Medicine in Sport outside the submitted work. HW reports being a minority shareholder of Uplanner BV and Replasia BV outside the submitted work.

**Patient and public involvement** Patients and/or the public were involved in the design, or conduct, or reporting, or dissemination plans of this research. Refer to the Patient and public involvement section for further details.

**Patient consent for publication** Not applicable.

**Ethics approval** This study involves human participants but Erasmus MC Medical Ethics Review Committee (MERC) exempted this study, because it uses previously collected observational data for which the participants had already given informed consent, and all cohort studies included in this consortium already had ethics Approval from their respective committees or boards. Participants gave informed consent to participate in the study before taking part.

**Provenance and peer review** Not commissioned; externally peer reviewed.

**Data availability statement** Data are available upon reasonable request. Data may be obtained from a third party and are not publicly available. We will provide a harmonised database containing all prospective data on hip OA. We encourage the use of data by third parties, although this is subject to approval by the steering committees of the World COACH consortium and the participating cohorts, as well as to legal boundaries regarding data ownership. To streamline the processing of third-party requests, we have developed a standardised data request form that can be distributed and reviewed uniformly. This will ensure consistency in the way data requests are handled within World COACH.

**ORCID iDs**
Michiel M A van Buuren http://orcid.org/0000-0001-7290-0145
Flavia Cicuttini http://orcid.org/0000-0002-8200-1618
David Felson http://orcid.org/0000-0002-2668-2447
John A Lynch http://orcid.org/0000-0003-3624-2741

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
