## [Reviewer comments · BMJ Open]

ARTICLE DETAILS

TITLE (PROVISIONAL)	Cohort profile: Worldwide Collaboration on OsteoArthritis prediction for the Hip (World COACH); an international consortium of prospective cohort studies with individual participant data on hip osteoarthritis.
AUTHORS	van Buuren, Michiel; Riedstra, Noortje; van den Berg, Myrthe; Boel, Fleur; Ahedi, Harbeer; Arbabi, Vahid; Arden, Nigel; Bierma-Zeinstra, Sita; Boer, Cindy; Cicuttini, Flavia; Cootes, Timothy; Crossley, Kay; Felson, David; Gielis, Willem Paul; Heerey, Joshua; Jones, Graeme; Kluzek, Stefan; Lane, Nancy; Lindner, Claudia; Lynch, John A.; Van Meurs, J; Mosler, Andrea; Nelson, Amanda; Nevitt, M; Oei, Edwin; Runhaar, Jos; Tang, Jinchi; Weinans, Harrie; Agricola, Rintje

VERSION 1 – REVIEW

REVIEWER	Gwynne-Jones, David P. University of Otago, Orthopaedic Surgery
REVIEW RETURNED	08-Aug-2023

GENERAL COMMENTS	This cohort profile describes the establishment of a consortium of researchers based on a literature review of cohort studies that include data and radiology of the hip. Nine prospective cohort studies were identified including 38,000 patients. The paper describes the literature search, recruitment of collaborators, funding, data harmonisation and goals of the consortium. The Introduction emphasized the global burden of hip osteoarthritis and the limitations of current treatment. They hope that an increase in the knowledge on the aetiology, pathophysiology and risk factors of hip OA may help future treatment strategies. Several risk factors are noted including age, gender, genetics, race and hip morphology although no comment has been made on obesity. The stated aim of the consortium was to better understand risk factors for the development or progression of hip osteoarthritis and to optimise and to automate methods for analysing radiological images of the hip. I think that there should be greater emphasis put in the Introduction on the morphological characteristics that may predispose to this such as dysplasia, cam and pincer impingement. There has been extensive literature on radiological measurements from plain x-rays and from 3-D imaging. I note that two of the authors have a software package designed to automate this which raises the question of conflict of interest. Description of the Cohorts A summary of the cohorts is presented in Table 1. These reveal a relatively heterogenous group of cohort studies ranging from
--

general population, osteoporotic elderly females, young Australian football players, and those patients recruited because of knee pain or osteoarthritis. Each study is described in a paragraph which adds little detail beyond that given in Table 1. No details are given of the data collected such as pain or outcome scores, blood tests, genetic testing or otherwise.

Data Harmonisation

Data harmonisation was discussed but this reads more as a study protocol than giving detail on the profile of the cohort. The section on data storage and processing is appropriate. Lines 277 - 279 states that the first steps regarding defining research questions and objectives have already been described. However the main objective of the study is only stated in line 339, namely the development of hip OA within the various follow up periods. They propose to define hip OA by radiological indices, clinically by pain or function indices or by a combination of these two. I think this needs to be clarified as part of the methods section. The main outcomes from the study are the development of new radiographic hip OA or progression of radiographic hip OA and pain scores.

Findings to Date

The only findings presented to date are summarised in Table 2 showing that 38,021 individuals have been included of whom only 34,00 have had baseline x-rays despite this being an inclusion criteria for the study. The mean age was 65.3 years and 71% were women. The following paragraph on the five separate work packages does not reflect findings but should be part of introduction or methods. No data has been presented to clarify the genetics work package. The fourth package includes clinical measures comprising physical examination, questionnaires, quality of life, blood and urine samples however it is unclear how many of these have been collected in each individual cohort study.

Strengths and Limitations

The main strength of the study is that it is pooled data from 38,000 individuals, some of whom are likely to have had early osteoarthritis. The second part of the strengths paragraph is an aspirational goal to develop a risk specific estimation. Lines 394 - 398 again refers to radiographic data analysis which should have been discussed in greater detail in the introduction.

I agree that there are limitations with predominantly cohorts from Australia, Europe and the USA. The cohorts do not reflect a general population as they include some patients with pre-existing hip symptoms, pre-existing knee osteoarthritis and are generally middle aged and elder. Interventions based on hip morphology would be more relevant to include a younger cohort of patients. 3-D imaging such as CT or MRI would provide much more useful data but I accept the greater utility of using plain radiographs.

Collaboration

Is all well written and relevant. Due attention has been paid to patient and public involvement.

Cohort profiles

I admit to having limited knowledge of cohort profiles. Using the notes from the BMJ Open: Cohort profiles should describe a large collaborative prospective study that identifies a group of participants and follows them for long periods.

	Most of these studies are historical. Many have stopped recruiting. They will usually be population based In this case many of the studies are a sub group of patients with varying pathologies rather than from the general population. I note that the average BMI was 27.4. This is certainly not representative of most patients presenting for total hip replacement where a BMI in excess of 30 is now the norm in many countries. Therefore I question how generalizable this cohort would be. Sufficient funding There appears to be sufficient funding to run the COACH study but the individual cohorts have been completed. BMJ Open publishes cohort profiles to provide information on a cohort's establishment that goes beyond what can reasonably be described in the methods section of a research paper. I do not think there is enough detail in the paper to publish it in this form. I think the establishment of this collaboration and the data so far reported could reasonably be included in the methods section of a results research paper. The table summarising the participants has little more than age, gender, BMI. It could be expanded to include details of other clinical measures, timing and percentage of subsequent radiological investigations and various other data collected. Systematic reviews will typically include much of this in their tables or supplementary tables. The original investigators in this group appear to welcome the wide use of data sets beyond their own group. The questions raised in the section on 'Why publish cohort profiles' pertaining to cohort studies I think also pertain to this cohort profile. It is not clear exactly how the patients were recruited. They clearly are not representative of the general population and no details were given on the questions used to gather information etc. Conclusion This cohort is based on retrospective use of previously collected data, much of which was collected without the goal of following hip osteoarthritis. There is a lot of extraneous detail on the selection of papers and development of this collaboration, rather than details of the cohorts. I find the format and structure of the paper confusing. The initial aims are quite clear, namely the identification of new hip OA or progression of disease by radiological and pain scores. Some base line data on this at recruitment of the patients would be useful. Although there are 38,000 patients in the study, only 34,000 had initial base line radiology. There is no indication of the incidence of hip OA in this large group. Therefore no idea of the power of the study can be gained. It is also unclear how the immediate benefits of the development of a patient specific risk estimation of developing hip osteoarthritis especially given the limited efficacy of current treatments. I applaud the underlying goals of the COACH consortium and recognise its potential to develop a large prospective data base of patients who have had various data collected including radiological examination. I feel that this cohort profile in its current form is not suitable for publication. It is a mixture of a cohort profile that lacks sufficient details to stand on its own and as a research protocol it also lacks detail. However it remains the editors decision whether this fulfils the requirement for publication.
--	---

REVIEWER	Appleyard, Tom
----------	----------------

	Keele University
REVIEW RETURNED	10-Oct-2023

GENERAL COMMENTS	This is a great collaborative project which must be commended and the consortium's article is well written. I have a few minor points that I believe would strengthen the article, and aid in the application of this project which will be its ultimate aim.  1. The background is clearly structured although I feel the argument regarding "Why Hip OA?" could be strengthened. There is clear articulation that hip and knee OA correspond with the greatest level of disability. However, a brief overview of prediction models to date, and how these predominantly focus on knee OA, would identify a rationale for focussing on hip OA. Furthermore, prediction models of hip OA to date have been limited to cohorts of patients from the Netherlands (as per 2022 systematic review; Appleyard et al), reinforcing need for collaboration/diversification. Finally, more specific figures regarding the burden of hip OA particularly would be beneficial in strengthening the background. 2. The paragraphs regarding cohort descriptives are well written. 3. More discussion is warranted regarding applicability of the harmonised data (within the harmonisation subsection or within potential limitations) and its implication for the various packages. For packages Two to Five (i.e. understanding effect of hip morphology, genetics and clinical markers on hip OA risk and ultimately a prediction model), further discussion is needed whether the cohorts are heterogenous to the point that the "noise" within the datasets results in no meaningful individualised prediction being possible. An obvious example of this includes inclusion of a dataset of sportspeople, that are likely to have a different pathology and natural history and may have post-traumatic arthritis. Whilst there may be predictors (occupation, previous trauma) that can account for this, I wonder whether its inclusion is beneficial to the wider pool of data. 4. The variation in possible diagnoses of hip OA requires expansion/justification. It is stated that cohorts predicting clinical, radiographic or combined hip OA were eligible, but again, has data been pooled that is effectively diagnosing differing conditions (or at least at differing timepoints/severity)? Can these outcomes be meaningfully harmonised to suggest patients across cohorts have reached the same endpoint? 5. Whilst I appreciate that cohorts for knee OA that provide pelvic radiographs at multiple timepoints are appropriate for inclusion, I think this could be more explicitly stated as to not confuse readers. 6. When discussing the potential inclusion of future studies, learning points from currently included cohorts could be addressed. Limitations mentioned in the background include the reliance on Western-world cohorts. However, other aspects noted within the aforementioned systematic review include the lack of social stratifiers (with race occasionally included), as well as a reliance on imaging. With imaging not featuring in NICE (UK) guidance or within ACR Clinical Criteria (USA), the reliance on imaging within cohorts merits highlighting for ultimate applicability to clinical practice.
--

	7. Minor point but line 447 "hip OA patients" should be changed to "patients with hip OA" to avoid possible labelling.
--	--

VERSION 1 – AUTHOR RESPONSE

Response to reviewer comments

Reviewer: 1

Dr. David P. Gwynne-Jones, University of Otago

General response:

Dear Dr. Gwynne-Jones,

We thank you for your effort in peer-reviewing our consortium profile manuscript, and we appreciate that you have clearly invested a lot of time in this extensive review.

In general, we would like to clarify the purpose of our cohort profile paper. The guidelines as found on https://bmjopen.bmj.com/pages/authors#cohort_profile state that a cohort profile should bridge the gap between a study protocol and a results paper. Because many projects in this consortium have just started or have yet to be started, we have little original results yet, so we leaned more towards a study protocol. This is also in line with other consortia profile papers as previously published in BMJ open (e.g. eye and vision consortium of UK Biobank, FINNPEC consortium, GECCO consortium, MULTITUDE consortium).

Consequently, in our paper we have made some conscious choices regarding what to describe and not to describe. Our goal with this paper is to describe how we identified the eligible cohorts with a formal systematic search and to provide a general overview of the consortium, its aims and work packages, and some universal methodological aspects that will otherwise need repetition in each following paper (such as the description of the cohort, radiographic protocols, etc.). As we believe the research possibilities are virtually endless, it is impossible to go in-depth on each of the planned research projects. Moreover, since a consortium is continuously a work in progress, we cannot yet answer all specific questions regarding methodology (e.g. about pain or outcome scores, blood tests, genetic testing, etc.), since these will be addressed when those projects are truly started. The same goes for data harmonisation: we described the general principles we will be using for data harmonisation, but are unable to describe this for every variable as this is ongoing work.

As we will be using radiographs to determine the presence/absence of hip OA, we felt it was useful to describe the radiographic protocols in detail, as well as the populations and in- and exclusion criteria of the included cohorts, so that we can refer to this cohort profile in all follow-up studies.

We have added our specific responses and the associated actions in the manuscript below under each paragraph of your comments (with line numbers referring to the version with tracked changes, options: all markup shown + revisions in balloons):

Comments to the Author:

This cohort profile describes the establishment of a consortium of researchers based on a literature review of cohort studies that include data and radiology of the hip. Nine prospective cohort studies were identified including 38,000 patients. The paper describes the literature search, recruitment of collaborators, funding, data harmonisation and goals of the consortium.

The Introduction emphasized the global burden of hip osteoarthritis and the limitations of current treatment. They hope that an increase in the knowledge on the aetiology, pathophysiology and risk factors of hip OA may help future treatment strategies. Several risk factors are noted including age, gender, genetics, race and hip morphology although no comment has been made on obesity. The

stated aim of the consortium was to better understand risk factors for the development or progression of hip osteoarthritis and to optimise and to automate methods for analysing radiological images of the hip. I think that there should be greater emphasis put in the Introduction on the morphological characteristics that may predispose to this such as dysplasia, cam and pincer impingement. There has been extensive literature on radiological measurements from plain x-rays and from 3-D imaging. I note that two of the authors have a software package designed to automate this which raises the question of conflict of interest.

Responses:

- Although the results on the association between obesity and development of hip OA (unlike knee OA) are conflicting, we agree with the reviewer that obesity should be listed as a potential risk factor.
→ Added obesity in the introduction in **line 119**.
- Recent evidence indeed points towards an important role for hip morphology in the development of OA. Although we dedicated a work package (1 out of 5 work packages) to hip morphology in our consortium, we initially did not specify hip morphology further, as our goal is to describe the consortium in general without an imbalance to a given work package. We adjusted this now to be more specific on the types of hip morphology we will be focusing on.
→ Added specific morphological risk factors in **lines 120-121**.
- We understand your concerns regarding two of our authors owning a software package that we use to automate statistical shape analysis, and this would certainly be a problem if there were any commercial benefits associated. However, BoneFinder® is freely available for non-commercial research purposes, and we plan to make any of the used scripts and models open-access for use by the scientific community as well.

Description of the Cohorts

A summary of the cohorts is presented in Table 1. These reveal a relatively heterogenous group of cohort studies ranging from general population, osteoporotic elderly females, young Australian football players, and those patients recruited because of knee pain or osteoarthritis. Each study is described in a paragraph which adds little detail beyond that given in Table 1. No details are given of the data collected such as pain or outcome scores, blood tests, genetic testing or otherwise.

Responses:

- We believe that the heterogeneity of populations within this consortium is actually a strength, since it accommodates subgroup analyses and allows more generalisable results. Further, for the eventual aim to develop an artificial intelligence-based personalised prediction model, clear international guidelines are still lacking but recent recommendations include that the real-world heterogeneity and diversity should be covered as much as possible (*de Hond et al. Guidelines and quality criteria for artificial intelligence-based prediction models in healthcare: a scoping review. npj Digit. Med. 5, 2 (2022). <https://doi.org/10.1038/s41746-021-00549-7>*). Of course we will investigate and account for heterogeneity in our analyses.
- As stated in the general comment, we had opted for providing only those data for each cohort that would be universally necessary for each of the work packages and future studies. Also, we currently cannot provide more details on large parts of other variables in our consortium, partly because the harmonization is still ongoing and they are too heterogeneous to concisely describe them (e.g. pain, biomarkers, genetics).
→ We do agree with the reviewer that these paragraphs could provide more details, and have added more information concerning the population, recruitment, and inclusion criteria, starting from **line 221**.

Data Harmonisation

Data harmonisation was discussed but this reads more as a study protocol than giving detail on the profile of the cohort. The section on data storage and processing is appropriate. Lines 277 - 279 states that the first steps regarding defining research questions and objectives have already been described. However the main objective of the study is only stated in line 339, namely the development of hip OA within the various follow up periods. They propose to define hip OA by radiological indices, clinically by pain or function indices or by a combination of these two. I think this needs to be clarified as part of the methods section. The main outcomes from the study are the development of new radiographic hip OA or progression of radiographic hip OA and pain scores.

Responses:

- As stated in the general comment, our interpretation of this cohort profile in the context of a consortium is more of a study protocol indeed, rather than going into a lot of details. The description of the cohorts, baseline characteristics, radiographic protocols, etc. will be needed in every methods section of follow-up papers; for which we can refer to this consortium profile. If a future study uses specific additional variables (genetics, biomarker, questionnaire data), those specific variables will be explained in detail in that paper or in an appendix.
- With the statement in lines 277-279 we were originally referring back to the aims of the consortium as stated in the introduction, but the reviewer is correct that this was unclear and not very elaborate.
→ We have restructured the manuscript so that objectives and work packages is now mentioned earlier (starting from **line 155**), which reads more logically.
- We have opted not to strictly define the outcomes (the definitions of hip OA) in this consortium profile, since the desired outcome may vary per future research project within this consortium, sometimes focusing on radiographic hip OA and sometimes on clinical hip OA, or a combination. Even within an apparently well-defined outcome such as radiographic hip OA, there is much heterogeneity (Kellgren-Lawrence, Croft, OARSI) and there is no golden standard in the literature yet. Within the Methodology work package we are focusing on this problem, and it is possible that a future consortium publication will focus merely on the definition of radiographic hip OA and/or validation of a new scoring method (AI-driven?).

Findings to Date

The only findings presented to date are summarised in Table 2 showing that 38,021 individuals have been included of whom only 34,000 have had baseline x-rays despite this being an inclusion criteria for the study. The mean age was 65.3 years and 71% were women. The following paragraph on the five separate work packages does not reflect findings but should be part of introduction or methods. No data has been presented to clarify the genetics work package. The fourth package includes clinical measures comprising physical examination, questionnaires, quality of life, blood and urine samples however it is unclear how many of these have been collected in each individual cohort study.

Responses:

- We thank the reviewer for his comment; we now clarified that the availability of consecutive X-rays was an inclusion criterion on cohort level but not on participant level (**lines 179-181**). We think it is critical to also include the participants with missing X-rays from the eligible cohorts, as it is vital to account for missing data and potential bias, particularly for the purpose of prediction modelling.
- We agree that the paragraph on work packages does not reflect findings. Although there is no methods section for this type of paper according to the author guidelines, we have restructured our manuscript so it will read more logically (see previous comment).
→ Work packages are now described in a different section starting **line 155**.
- In line with previous comments, there is currently no detailed data yet from the genetics and clinical measures work packages. We know which cohorts have these type of data available, but the identification of the exact numbers and the way to harmonise these variables is still a work in progress.

Strengths and Limitations

The main strength of the study is that it is pooled data from 38,000 individuals, some of whom are likely to have had early osteoarthritis. The second part of the strengths paragraph is an aspirational goal to develop a risk specific estimation. Lines 394 - 398 again refers to radiographic data analysis which should have been discussed in greater detail in the introduction.

Response:

- We agree with the reviewer that this knowledge gap should have been mentioned in the introduction.
→ We have added some additional information regarding the developments in radiological image processing and its implications for the consortium (**lines 126-134**).

I agree that there are limitations with predominantly cohorts from Australia, Europe and the USA. The cohorts do not reflect a general population as they include some patients with pre-existing

hip symptoms, pre-existing knee osteoarthritis and are generally middle aged and elder. Interventions based on hip morphology would be more relevant to include a younger cohort of patients. 3-D imaging such as CT or MRI would provide much more useful data but I accept the greater utility of using plain radiographs.

Response:

- Some cohorts included their participants from the general population while others included participants based on certain characteristics that probably increase their chance of developing OA, such as hip pain, knee OA and/or a higher BMI. This mix of participants from the general population and participants more prone to develop hip OA is ideal for studying subgroups. In our analyses, we will take into account from what type of population the participants come from. Almost all cohorts included participants from the age of 45 years and above. This captures the majority of people from the general population that develop hip OA. We acknowledge that we cannot generalise our findings to a younger population as people who develop OA at a younger age are also likely to have other underlying causes or risk factors. We can partly use the FORCe cohort for hypothesis generation on early-onset hip OA, as FORCe did include younger participants (18-50 years), but given the smaller size of this cohort we will not be able to externally validate these findings. Therefore, we will be focusing on the group of people aged >45 years, which is also the vast majority in terms of hip OA incidence, and this has now been added as a discussion point (**lines 603-606**).
- We agree that for interventions based on hip morphology it would be beneficial to include younger participants. However, this consortium is not focused on interventions, but merely on aetiology and risk factors based on epidemiological data. Therefore, we only included observational cohort studies and not intervention studies, which are beyond the scope of our consortium.
- We also agree that 3D imaging would provide more extensive data and can be beneficial for certain research questions, but this is only available for a limited number of participants. Study results gained from 3D imaging may also limit real-world application since OA patients rarely have 3D imaging done, while plain radiography is amply done. Nevertheless, we are certainly planning projects with the participants that do have 3D imaging available at least within the Methodology and Hip Morphology work packages. Still, this is a relevant point raised by the reviewer, hence it was already elaborated on in the limitations section.

Collaboration

Is all well written and relevant. Due attention has been paid to patient and public involvement.

Cohort profiles

I admit to having limited knowledge of cohort profiles. Using the notes from the BMJ Open: Cohort profiles should describe a large collaborative prospective study that identifies a group of participants and follows them for long periods. Most of these studies are historical. Many have stopped recruiting.

Response:

- We acknowledge that 6 of the included studies have already completed their follow-up, however, based on previous consortium profiles published as such in BMJ open, we have interpreted this differently in the context of a consortium. Almost by nature a consortium is not prospective in their data collection but rather 're-uses' data in order to harmonise and pool the data in an 'individual participant data meta-analysis' fashion.

They will usually be population based

In this case many of the studies are a sub group of patients with varying pathologies rather than from the general population. I note that the average BMI was 27.4. This is certainly not representative of most patients presenting for total hip replacement where a BMI in excess of 30 is now the norm in many countries. Therefore I question how generalizable this cohort would be.

Response:

- It is true that some of the studies have recruited subgroups of individuals with varying pathologies or other prerequisites (i.e., CHECK, FORCe, MOST, OAI), which is useful for subgroup analysis. However, Chingford, JoCo, the Rotterdam Study, SOF and TASOAC have all recruited participants from the general population, albeit with an age restriction. In order to

make the latter more clear in the manuscript, we have added more information to the cohort paragraphs as previously stated.

- We do not agree with the statement that this population is not representative of most patients presenting for total hip replacement. Looking at the 2022 annual report by the Dutch Arthroplasty Register for example, 64% of OA-related total hip replacements are in women, mean age is 70, and mean BMI is 27.4 (with 75% falling in the BMI 18.5-30 category). This is more or less the average demographic of our consortium. There might be differences across populations indeed, which is why it is so important to have the statistical power for subgroup analyses, for example based on BMI (e.g. in MOST, OAI and JoCo, where mean BMI is 29-30), or based on other characteristics.

Sufficient funding

There appears to be sufficient funding to run the COACH study but the individual cohorts have been completed.

BMJ Open publishes cohort profiles to provide information on a cohort's establishment that goes beyond what can reasonably be described in the methods section of a research paper.

I do not think there is enough detail in the paper to publish it in this form. I think the establishment of this collaboration and the data so far reported could reasonably be included in the methods section of a results research paper. The table summarising the participants has little more than age, gender, BMI. It could be expanded to include details of other clinical measures, timing and percentage of subsequent radiological investigations and various other data collected. Systematic reviews will typically include much of this in their tables or supplementary tables.

Response:

- We have expanded the relevant sections with some additional details as stated above. We would like to stress again that not all details from each variable available can be described in this general overview paper, especially not details about the work packages and projects that are still starting up or have not even started yet. We would also like to point out that the relevant sections of this manuscript count over 5,000 words, which is already beyond the conventional *total* word count for original research, let alone the methods section.

The original investigators in this group appear to welcome the wide use of data sets beyond their own group.

The questions raised in the section on 'Why publish cohort profiles' pertaining to cohort studies I think also pertain to this cohort profile.

It is not clear exactly how the patients were recruited. They clearly are not representative of the general population and no details were given on the questions used to gather information etc.

Response:

- The process of identification and recruitment of cohorts for the consortium itself has been done with a formal systematic review of available literature, as described. Following your suggestions, we have expanded the paragraphs containing the cohort descriptions with the recruitment process of each cohort. No detailed information could be given on the questions used to gather information, since there are dozens to hundreds of questionnaires per cohort, with a mix of home visits, telephone calls, clinic visits, sometimes varying per follow-up moment. Some of this information can be found in the original cohort's study protocols (which are cited in each relevant paragraph), and the important information will be included in future manuscripts that use data from those respective questionnaires.

Conclusion

This cohort is based on retrospective use of previously collected data, much of which was collected without the goal of following hip osteoarthritis. There is a lot of extraneous detail on the selection of papers and development of this collaboration, rather than details of the cohorts.

I find the format and structure of the paper confusing. The initial aims are quite clear, namely the identification of new hip OA or progression of disease by radiological and pain scores. Some base line data on this at recruitment of the patients would be useful. Although there are 38,000 patients in the study, only 34,000 had initial base line radiology. There is no indication of the incidence of hip OA in this large group. Therefore no idea of the power of the study can be gained. It is also unclear

how the immediate benefits of the development of a patient specific risk estimation of developing hip osteoarthritis especially given the limited efficacy of current treatments.

Response:

- We believe most of the issues raised have been addressed in previous responses. We added baseline data and described the recruitment processes of the cohorts. We explained the importance to also include participants with missing or unavailable imaging data. The added **Supplemental Table S2** gives an estimate of the actual prevalence/incidence of radiographic hip OA in the consortium, using the original cohort's most easily presentable scores.
→ We have added **Table 3 and Supplemental Tables S1-2** and provided some more details in **lines 541-557**. As we do not have a harmonised RHOA variable yet, this outcome may change in future projects.
- The limited efficacy of current treatments is also due to the current one-size-fits-all approach. By identifying risk factors on an individual level, a more tailored approach might be possible. For example, in certain overweight people BMI might be a risk factor for hip OA development (for example due to an underlying genetic background) while in other overweight people other factors than BMI might play a role (for example hip morphology). Also; identifying people with a certain morphological risk factor (cam morphology or dysplasia) that will develop hip OA with a higher likelihood (based on symptoms, clinical examination findings, genetics, etc.) than others with the same hip morphology, will allow more targeted treatment strategies (either surgical or conservative) that possibly prevent or delay the onset of hip OA. Finally, we might be able to identify new, currently unknown risk factors based on which potential new treatment strategies could be developed in the future.

I applaud the underlying goals of the COACH consortium and recognise its potential to develop a large prospective data base of patients who have had various data collected including radiological examination. I feel that this cohort profile in its current form is not suitable for publication. It is a mixture of a cohort profile that lacks sufficient details to stand on its own and as a research protocol it also lacks detail. However it remains the editors decision whether this fulfils the requirement for publication.

Response:

- We thank you for supporting our cause and recognising its potential, and we especially thank you for your useful comments and suggestions that have hopefully improved this manuscript and made it suitable for publication.

Reviewer: 2

Dr. Tom Appleyard, Keele University, Health Education England North East

General comment:

Dear Dr. Appleyard,

We thank you for your relevant and insightful comments. We have added our responses and the associated actions in the manuscript below under each paragraph of your comments (with line numbers referring to the version with tracked changes, options: all markup shown + revisions in balloons):

Comments to the Author:

This is a great collaborative project which must be commended and the consortium's article is well written. I have a few minor points that I believe would strengthen the article, and aid in the application of this project which will be its ultimate aim.

1. The background is clearly structured although I feel the argument regarding "Why Hip OA?" could be strengthened. There is clear articulation that hip and knee OA correspond with the greatest level of disability. However, a brief overview of prediction models to date, and how these predominantly focus on knee OA, would identify a rationale for focussing on hip OA. Furthermore, prediction models of hip OA to date have been limited to cohorts of patients from the Netherlands (as per 2022 systematic review; Appleyard et al), reinforcing need for collaboration/diversification. Finally, more specific figures

regarding the burden of hip OA particularly would be beneficial in strengthening the background.

Response:

- Thank you for your suggestions on how to strengthen this argument.
→ We have elaborated on the reason for investigating hip OA specifically in **lines 108-112** and **lines 126-134**.

2. The paragraphs regarding cohort descriptives are well written.

Response:

- Thank you. As per reviewer 1's suggestion, we have elaborated further on the cohort descriptions in the current version, particularly on their recruitment strategies.

3. More discussion is warranted regarding applicability of the harmonised data (within the harmonisation subsection or within potential limitations) and its implication for the various packages. For packages Two to Five (i.e. understanding effect of hip morphology, genetics and clinical markers on hip OA risk and ultimately a prediction model), further discussion is needed whether the cohorts are heterogeneous to the point that the "noise" within the datasets results in no meaningful individualised prediction being possible. An obvious example of this includes inclusion of a dataset of sportspeople, that are likely to have a different pathology and natural history and may have post-traumatic arthritis. Whilst there may be predictors (occupation, previous trauma) that can account for this, I wonder whether its inclusion is beneficial to the wider pool of data.

Response:

- Thank you for this suggestion, we agree that it is important to discuss the applicability of harmonised data in the real world within the consortium profile paper, including a discussion on heterogeneity. It is probable that there is a large amount of heterogeneity among the included cohorts. We have tried to filter for this during study selection, and will try to add as much predictors as possible in any future statistical model, to optimally correct for heterogeneity.
→ We have added this to the discussion in lines **596-606**.
- Regarding the FORCe cohort, we agree that this may be a different population with different predictors. These young participants who possibly have traumatic risk factors may give some insight into the development of early-onset hip OA, possibly through different pathways. That's why we are planning to investigate multiple research questions and do subgroup analyses. FORCe participants may often end up being left out of the large consortium analyses, but we still believe there is benefit especially as their follow-up extends. On top of that, their inclusion of 3D imaging (MRI) can be very valuable in some separate analyses, including analyses for the Methodology work package (e.g. comparison of apparent hip morphology on 2D vs 3D imaging).

4. The variation in possible diagnoses of hip OA requires expansion/justification. It is stated that cohorts predicting clinical, radiographic or combined hip OA were eligible, but again, has data been pooled that is effectively diagnosing differing conditions (or at least at differing timepoints/severity)? Can these outcomes be meaningfully harmonised to suggest patients across cohorts have reached the same endpoint?

Response:

- This is a very good question and one we have been struggling with within our group. In the current stage of the consortium, we have entered all available data in a database, and we are still discussing harmonisation options. We do agree that it is impossible to harmonise data that is effectively diagnosing different conditions, so for one we will not harmonise clinical endpoints with radiographic endpoints. Fortunately, all included cohorts have strictly defined radiographic OA as an outcome, which will also be our main outcome and which is in our opinion the 'easiest' to harmonise (still not easy). If corresponding clinical data is available separately from radiographic scores, and we see a viable option to combine this into a combined endpoint, we will try this in a separate validation project before using it as an actual outcome. We are currently investigating the options regarding different time points and severity of the outcomes with statisticians. This may include ordinal logistic regression (to model severity) and/or mixed models that include 'time to event' as a factor. This is not

included in the paper as the author guidelines advise not reporting detailed statistical plans and as this is still a work in progress.

5. Whilst I appreciate that cohorts for knee OA that provide pelvic radiographs at multiple timepoints are appropriate for inclusion, I think this could be more explicitly stated as to not confuse readers.

Response:

- We agree that this may not be immediately clear to the general reader, so we have explained this more explicitly in both of the relevant paragraphs.

6. When discussing the potential inclusion of future studies, learning points from currently included cohorts could be addressed. Limitations mentioned in the background include the reliance on Western-world cohorts. However, other aspects noted within the aforementioned systematic review include the lack of social stratifiers (with race occasionally included), as well as a reliance on imaging. With imaging not featuring in NICE (UK) guidance or within ACR Clinical Criteria (USA), the reliance on imaging within cohorts merits highlighting for ultimate applicability to clinical practice.

Response:

- Thank you for bringing this up. We are working on each of these point currently, but did not yet mention them yet in the discussion. We have been trying to include an Asian cohort, but this is still not concluded.
- A lot of the included cohort studies do have social stratifiers in their data, but the heterogeneity is substantial and we are still figuring out if and how we can use those in our analyses. If we can meaningfully harmonize them, we will.
→ We have now briefly mentioned this in **lines 559-564**.
- The lack of need for radiographic imaging in diagnosing hip OA is something we are aware of, and have discussed within our group. We aim to build prediction models both with and without imaging data (including hip morphology as a predictor and radiographic hip OA as outcome) to find out what imaging adds. Perhaps this may also lead to more insight into which patients should actually be sent for imaging by their GP, and which patients could just start with the conservative treatment first based on a clinical diagnosis of hip OA (as the guidelines say).
→ We have now added some clarification on this in **lines 612-617** and **621-623**.

7. Minor point but line 447 "hip OA patients" should be changed to "patients with hip OA" to avoid possible labelling.

Response:

- We agree and have changed this accordingly in both instances.

Reviewer: 1

Competing interests of Reviewer: None

Reviewer: 2

Competing interests of Reviewer: No competing interests applicable